# Metabolic Comorbidities and Cardiovascular Disease in Pediatric Psoriasis: A Narrative Review

**DOI:** 10.3390/healthcare10071190

**Published:** 2022-06-25

**Authors:** Andrea Marani, Giulio Rizzetto, Giulia Radi, Elisa Molinelli, Irene Capodaglio, Annamaria Offidani, Oriana Simonetti

**Affiliations:** 1Clinic of Dermatology, Department of Clinical and Molecular Sciences, Polytechnic University of Marche, 60121 Ancona, Italy; andreamarani.med@yahoo.com (A.M.); grizzetto92@hotmail.com (G.R.); radigiu1@gmail.com (G.R.); molinelli.elisa@gmail.com (E.M.); annamaria.offidani@ospedaliriuniti.mache.it (A.O.); 2Hospital Cardiology and UTIC, Reunited Hospitals Torrette of Ancona, 60126 Torrette, Italy; irene.capodaglio@ospedaliriuniti.marche.it

**Keywords:** psoriasis, children, comorbidities, metabolism, obesity, diabetes, lipids, lipoprotein, peroxidation

## Abstract

Psoriasis vulgaris is a common inflammatory, immune mediated, chronic recurrent dermatosis. Psoriasis is also a systemic inflammatory disease, associated with numerous comorbidities, particularly metabolic ones. Here, we summarize and discuss, in a narrative review, the current knowledge about the metabolic comorbidities in psoriatic children. Obesity, insulin resistance, diabetes, cardiovascular disease, and dyslipidemia are identified as the main comorbidities in psoriatic children. In conclusion, dermatologists should be aware of the metabolic comorbidities in children with psoriasis, modulating the therapeutic approach according to the patient’s clinical condition.

Psoriasis vulgaris is a common inflammatory, immune mediated, chronic recurrent dermatosis caused by the interplay between multiple genetic and environmental risk factors including vascular ones [1,2]. Today, psoriasis is known to not only be a cutaneous, but also a systemic inflammatory disease that is associated with numerous comorbidities, particularly metabolic ones [3]. Psoriasis begins in childhood in almost one-third of cases and affects 0.05% to 2.20% of children. Pediatric psoriasis is less common than adult psoriasis, and metabolic comorbidities of psoriasis are also uncommon in children than in adults [4]. However, recent studies have begun to demonstrate the association of pediatric psoriasis with systemic comorbidities, particularly metabolic, as is the case in adults. Metabolic comorbidities in pediatric psoriasis are still often overlooked and can have an impact on the patient’s health, so appropriate monitoring and management are desirable [4].

We decided to perform a narrative review of the literature because there is a lack of an easy-to-read overview of the metabolic comorbidities of pediatric psoriasis. We searched PubMed for the following terms: pediatric psoriasis, metabolic comorbidities, diabetes, obesity, dyslipidemia, hypertension, heart disease, and metabolic syndrome. English-language studies were preferred, with more emphasis on the latest evidence for each topic.

## 1. Obesity

Several studies have shown an association between obesity and psoriasis in children. Obese children have an increased risk of developing psoriasis [5,6]. Psoriatic children evaluated in an important international cross-sectional multicenter study globally had excess adiposity (estimated by body mass index percentile) and increased central adiposity (estimated by the waist circumference percentile and waist to height ratio) regardless of the psoriasis severity. A recent study that recruited 104 pediatric patients with psoriasis also found a significantly higher prevalence of central obesity in these subjects compared to the control group [7]. However, some correlation was observed between obesity severity and psoriasis severity. In the same study in fact, the odds ratio (95% CI) of obesity (body mass index >95th percentile) overall in psoriatic children versus the controls was 4.29 and was higher when associated with severe psoriasis; waist circumference above the 90th percentile occurred in 9.3% in the control, 14.0% of the mild psoriasis, and 21.2% of the severe psoriasis participants [6]. A recent systematic meta-analysis, instead, showed that the pediatric patient’s degree of obesity did not correlate with the severity of their psoriasis [8]. However, it remains unclear whether psoriasis or obesity comes first, although it appears more likely that obesity is a risk factor for the development of psoriasis than the other way around. Recent data have pointed out that children diagnosed with psoriasis are more likely to be obese before the diagnosis with respect to children without psoriasis, and, in those who are not obese at the time of diagnosis are likely to develop obesity after diagnosis as children without psoriasis, supporting the notion that obesity is a risk factor for psoriasis development, rather than the reverse [9,10]. It is also unclear, according to the current evidence, as to the mechanism by which overweight and obesity contribute to the onset of psoriasis and its severity. Psoriasis is associated with low-grade systemic inflammation supported by a variety of cytokines, chemokines, and other inflammatory mediators [11]. It is interesting that pediatric psoriasis patients exhibited a distinct difference in the expression of interleukin 17 (IL-17) and IL-22 compared to that of the healthy pediatric controls and adult psoriatic patients. Adipose tissue could be a source of low-grade inflammation in this scenario. According to recent evidence, leptin may play a key role in the relationship between obesity and psoriasis. This molecule, produced by adipose tissue and whose production is increased in the overweight patient, would be able to alter the subpopulations of immune cells, dysregulation that would be able to induce psoriasis on the skin [12]. Finally, it is not yet clear whether the weight reduction will have a lasting effect on the severity of psoriasis in the pediatric subpopulation [13].

## 2. Insulin Resistance and Diabetes

Insulin resistance is definable as the inability of cells to respond to the action of insulin produced by the pancreas. It is a condition that can lead to type 2 diabetes mellitus. The molecular mechanisms underlying insulin resistance are very complex and have not yet clarified, but a central role is once again played by adipose tissue. Many studies have reported a correlation between psoriasis and insulin resistance in adults, but there is an extreme lack of studies regarding the pediatric population [14]. A meta-analysis conducted by Pietrzak et al. [15] involving a total of 965 children found that the mean level of fasting glucose in children with psoriasis was 5.75 mg/dL, higher than in the healthy controls. Another interesting study was recently published by Caroppo et al. [16]. A total of 60 pediatric psoriatic patients were recruited, and insulin resistance was estimated by the HOMA-IR index (homeostatic model assessment for insulin resistance), a tool that also seems to be useful to estimate the cardiometabolic risk in children and adolescents. Insulin resistance was considered altered when the homeostatic model assessment (HOMA-IR) for insulin resistance was in the ≥90^th^ sex- and age-specific percentile and HOMA 2-IR was >1.8. The authors found the presence of insulin resistance in 16 out of the 60 patients recruited (27%), and all but four were overweight or obese (*n* = 12; 75%), and all but two had central obesity (*n* = 14; 88%), underlying the importance of evaluating the waist to height ratio in all children with psoriasis [17]. This also highlights the close relationship between obesity, insulin resistance, and psoriasis. The same authors investigated the prevalence of psoriasis in a court of pediatric patients with type 1 diabetes mellitus. The prevalence of psoriasis in these patients was four times higher than in the general population. The authors also observed the clinical onset of psoriasis was concurrent or subsequent to the diagnosis of diabetes. Moreover, the mean blood glycated hemoglobin level was slightly higher at the time of psoriasis onset than at the time of the visit, suggesting a potential role of hyperglycemia in the onset of pediatric psoriasis [17,18]. Regarding the correlation between type 2 diabetes mellitus and pediatric psoriasis, the extensive meta-analysis conducted by Phan et al. [8] found a statistically significant association. Overall, we can state that current scientific evidence suggests that there is an association between pediatric psoriasis and glucose dysmetabolism, but the literature is not exhaustive, especially regarding pathogenesis and clinical management.

## 3. Cardiovascular Diseases

The results of studies concerning the relationship between pediatric psoriasis and cardiovascular disease are contradictory and still have to be confirmed, although, according to current evidence, it can be stated that pediatric psoriasis has a significant association with several cardiometabolic comorbidities. With regard to hypertension, the systematic meta-analysis by Phan et al. [8] observed a statistically significant association of hypertension with pediatric psoriasis. Carobbo et al. [16] found hypertension, in the single-center study, in eight out of 60 patients (13% of the entire cohort). In contrast, in a cross-sectional study by Kelati et al. [19] involving 84 children with psoriasis and comorbidities, no cases of arterial hypertension were detected. In the meta-analysis performed by Pietrzak et al. [15], similarly, there was no difference in the systolic and diastolic blood pressure between the group of psoriatic children and the control group. Pediatric psoriasis is significantly associated with ischemic heart disease and heart failure [8]. An analysis conducted by Kwa et al. [20] on 4,884,448 hospitalized psoriatic children in the United States revealed that patients appeared to be at greater risk of metabolic cardiovascular comorbidities compared to valvular heart disease and arrhythmias. This finding may be interpreted in light of it having been observed in adult psoriasis, where a common pathogenesis has been demonstrated between cardiometabolic comorbidity and psoriatic disease, probably more than exists between valvulopathy and arrhythmia and psoriasis [21] However, the frequency of these increased cardiometabolic comorbidities would not be high in pediatric psoriatic patients [22]. Cardiovascular diseases are characterized by major alterations in lipid metabolism, and in pediatric psoriatic patients, it appears to be altered both quantitatively and qualitatively. An association between hyperlipidemia and pediatric psoriasis appeared to be statistically demonstrated by the largest and most recent meta-analyses [17]. From the perspective of quantitative lipid composition, Phan et al. [8] found no statistically significant difference between the LDL, HDL, triglyceride, and total cholesterol levels, considered individually, in pediatric psoriatic patients compared with the controls in contrast to other published studies in which strongly lower HDL values in psoriatic children were shown compared with the controls and normal triglycerides, but at the threshold of the statistically significant difference [18]. Similar results were found in the prospective study by Caroppo et al. [16,17], where 25% of their recruited patients had either high triglyceride values or low HDL values, and 18% had high total cholesterol values and 12% had high LDL values. In psoriatic children, a significant decrease in the apo-protein content in all lipoprotein fractions [23] and an increase in cholesterol associated with HDL, LDL, and VLDL have been observed [24]. Wynnis et al. [25] also found an increased atherogenic cardiometabolic risk in psoriatic children, with lipoprotein dysfunction (higher apolipoprotein B concentration, reduced HDL, and reduced cholesterol efflux capacity). Finally, a recent study showed an increase in the oxidative stress of serum lipoproteins related to a significant increase in the expression and activity of myeloperoxidase and a significant reduction in the activity of paraoxonase-1 [26].

## 4. Discussion

Metabolic syndrome is a set of clinical conditions, which, in this case, increases the risk of developing cardiac, vascular events, and diabetes. However, if in the adult the diagnosis of metabolic syndrome is defined by very precise criteria, this is not possible, for now, in the pediatric psoriatic patient, mainly due to the lack of precise age-related reference values in pediatric age, although an attempt has been made starting with the IDEFICS study, which developed a quantitative score to evaluate the metabolic syndrome using reference standards obtained in European children [27].

However, according to current scientific evidence, we can state that there is a higher prevalence of metabolic syndrome in children with psoriasis (i.e., a significant association with obesity, increased waist circumference, insulin resistance and diabetes, hypertension, dyslipidemia, and even cardiovascular disease). Dermatologists and pediatricians should be aware of this and take them into account at the time of the patient’s history, examination, and follow-up, with the aim of intercepting these comorbidities and preventing their progression. However, it seems clear that there are still contradictory results emerging from various studies that need to be clarified. Precise age-related criteria to define the metabolic syndrome in the pediatric patient, and therefore also in the pediatric psoriatic patient, are still lacking as well as precise guidelines for the management of these patients with the possibly intercepted comorbidities. An expert consensus document was produced [4], but with the strength of scientific recommendation C, which is a low level of scientific recommendation. Therefore, further studies are needed that focus not only on the statistically significant association between the comorbidities and pediatric psoriasis, but also on the pathogenesis of the disease, on the clinical management of comorbidities, and on the use of systemic drugs, taking into account the presence of comorbidities. In fact, the use of topical steroids in children is often related to therapeutic failure due to the phenomenon of corticophobia [28]. Systemic therapy must, in some cases, be considered when evaluating whether both the drugs may be contraindicated (acitretin, cyclosporine) or vice versa indicated such as certain biotechnological drugs. The latter may have the potential to reduce the impact of cardiometabolic comorbidity in psoriatic disease [29].

## 5. Conclusions

In conclusion, dermatologists should be aware of the metabolic comorbidities in children with psoriasis, modulating the therapeutic approach according to the patient’s clinical condition. In our opinion, topical therapy should be considered as the first-line treatment in cases of limited disease (Table 1). When systemic therapy is required, the effects of drugs on the metabolic comorbidities must be evaluated. Specifically, BMI screening, fasting serum glucose, universal lipid screening, and hypertension screening should be conducted for psoriatic children at risk of overweight or obesity and implementing lifestyle modifications with the support of a nutritionist to prevent their progression.

Among the conventional systemic drugs, acitretin, cyclosporine, and methotrexate should be avoided in the cases of obesity and dyslipidemia, the latter because it is potentially associated with an increased hepatotoxic risk in patients with obesity and diabetes. Fumaric acid esters and apremilast do not appear to have contraindications related to metabolic complications, but their use is still limited in the pediatric population [30].

According to our opinion, we suggest considering biological therapy as a first-line alternative in the event of topical therapy or phototherapy failure in pediatric psoriatic patient with a high risk of metabolic comorbidities. In particular, TNF alpha inhibitors (etanercept and adalimumab), IL-17 inhibitors (ixekizumab, secukinumab), and IL-12/23 inhibitors (ustekinumab) are available for pediatric use [30] and do not appear to have any interactions from the point of view of psoriasis metabolic complications.

## Figures and Tables

**Table 1 healthcare-10-01190-t001:** A summary of the metabolic comorbidities associated with pediatric psoriasis.

Metabolic Comorbidity Associated with Pediatric Psoriasis	Epidemiological Impact	Referral Study
** *Et Obesity* **	Obese children have an increased risk of developing psoriasis.	Augustin, M. et al. [5]
	Psoriatic children have excess adiposity and increased central adiposity regardless of psoriasis severity.	Paller, A.S. et al. [6]
	Pediatric patient’s degree of obesity does not correlate with the severity of psoriasis.	Phan, K. et al. [8]
	Children diagnosed with psoriasis are more likely to be obese before the diagnosis, with respect to children without psoriasis, and, in those who are not obese at the time of diagnosis are likely to develop obesity after diagnosis as children without psoriasis.	Becker, L. et al. [10]
	It is not yet clear whether the weight reduction will have a lasting effect on the severity of psoriasis in the pediatric subpopulation.	Gutmark-Little, I. et al. [14]
** *Insulin resistance* **	Mean level of fasting glucose in children with psoriasis was 5.75 mg/dL higher than in healthy controls.	Pietrzak, A. et al. [15]
	Presence of insulin resistance in 16 of 60 patients recruited (27%) into the study, and all but 4 were overweight or obese (*n* = 12; 75%) and all but 2 had central obesity (*n* = 14; 88%).	Caroppo, F. et al. [16]
** *Diabetes* **	Prevalence of psoriasis in patients with type 1 diabetes were 4 times higher than in the general population.	Caroppo, F. et al. [16]
	The clinical onset of psoriasis was concurrent or subsequent to the diagnosis of diabetes. Moreover, the mean blood glycated hemoglobin level was slightly higher at the time of psoriasis onset than at the time of the visit.	Caroppo, F. et al. [17]
	There is a statistically significant association between type 2 diabetes mellitus and pediatric psoriasis.	Phan, K. et al. [8]
** *Cardiovascular disease* **	There is a statistically significant association between hypertension and pediatric psoriasis.	Phan, K. et al. [8]
	Hypertension was found, in the single-center study, in 8 out of 60 patients (13% of the entire cohort).	Caroppo, F. et al. [16]
	In a cross-sectional study involving 84 children with psoriasis and comorbidities, no cases of arterial hypertension were detected.	Kelati, A. et al. [19]
	There was no difference in systolic and diastolic blood pressure between the group of psoriatic children and the control group.	Pietrzak, A. et al. [15]
	An analysis conducted on 4,884,448 hospitalized psoriatic children revealed that patients appear to be at greater risk of metabolic cardiovascular comorbidities compared to valvular heart disease and arrhythmias. However, the frequency of these increased cardiometabolic comorbidities would not be high in pediatric psoriatic patients.	Kwa, L. et al. [20]
	There was no statistically significant difference between the LDL, HDL, triglyceride, and total cholesterol levels, considered individually, in pediatric psoriatic patients compared with the controls.	Phan, K. et al. [8]
	25% of recruited patients (court of 60 patients) had either high triglyceride values or low HDL values, and 18% had high total cholesterol values and 12% had high LDL values.	Caroppo, F. et al. [16]
	Increased atherogenic cardiometabolic risk was detected in psoriatic children, with lipoprotein dysfunction (higher apolipoprotein B concentration, reduced HDL, and reduced cholesterol efflux capacity.	Tom, W.L. et al. [25]

## Data Availability

Not applicable.

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
