# Peer review of "Metabolic Comorbidities and Cardiovascular Disease in Pediatric Psoriasis: A Narrative Review"

_healthcare, 2022, doi:10.3390/healthcare10071190_

Round 1

Reviewer 1 Report

The paper presents a review of selected literature concerning metabolic comorbidities in pediatric psoriasis. The topic is interesting and has high clinical value. The introduction does not provide sufficient background about the psoriasis epidemiology and pathophysiology in children. The methodology of the paper is not described. It should be described what kind of comorbidities were taken in the analysis and explain why. It should be also described, what criteria were used in the literature selection. There is no many data concerning metabolic comorbidities in pediatric psoriasis that is why I found the paper valuable, because it can increase the awareness of dermatologists and pediatricians in this field.

Author Response

The paper presents a review of selected literature concerning metabolic comorbidities in pediatric psoriasis. The topic is interesting and has high clinical value.

The introduction does not provide sufficient background about the psoriasis epidemiology and pathophysiology in children.

Done

The methodology of the paper is not described. It should be described what kind of comorbidities were taken in the analysis and explain why. It should be also described, what criteria were used in the literature selection. There is no many data concerning metabolic comorbidities in pediatric psoriasis that is why I found the paper valuable, because it can increase the awareness of dermatologists and pediatricians in this field.

We searched PubMed for the following terms: pediatric psoriasis, metabolic comorbidities, diabetes, obesity, dyslipidemia, hypertension, heart disease, metabolic syndrome. English-language studies were preferred, with more emphasis on the latest evidence for each topic.

Reviewer 2 Report

Authors wrote a narrative review about pediatric psoriasis and comorbidities. The paper is not poorly written, but I think it should  be improved a bit. There are also a number of mistakes that need to be corrected. The authors have not presented any table in their manuscript, so I suggest making a table with comorbidities, and their prevalence. In addition to the comorbidities mentioned by the authors, it would be good to list rheumatoid arthritis and Crohn's disease, which also occur in pediatric psoriasis. Furthermore, since the authors in lines 51-53 mentioned cytokines it would be good to mention the difference in cytokines in children and adults in psoriasis. Pediatric psoriasis patients exhibited a distinct difference in the expression of interleukin 17 (IL-17) and IL-22 compared to that of healthy pediatric controls and adult psoriasis patients.

Other corrections to be made are as follows:

1. References in the text are not written according to the instructions. References should be written everywhere in square brackets.

2. Versus (vs) should be written in italic because it comes from a Latin word, and all Latin words are written in italic

3. Lines 41, 64, 65, 93, 107, 144 have too many spaces. Pass and correct the excess space in the text.

4. The names of the authors are not written in italic. Only collaborators are written in italics (et al.). Please revise the whole paper.

5. Also when the author's name is mentioned in the text, his name must be followed by a literary citation. Please revise the whole paper.

6. Lines 66-70 connect both sentences because the first one seems incomplete on its own.

7. Line 96 Pietrzak and coll should be Pietrzak et al. – please correct it.

8. Line 105 missing a dot at the end of the sentence.

9. Line 111 Phan and colleagues replace with Phan et al.

10. Lines 113-114 rewrite the text in black

11. Line 145 recommendation C - what is recommendation C, is it a mistake?

12. Line 150 the capital letter is missing at the beginning of the sentence.  A systemic therapy…

Author Response

Authors wrote a narrative review about pediatric psoriasis and comorbidities. The paper is not poorly written, but I think it should  be improved a bit. There are also a number of mistakes that need to be corrected. The authors have not presented any table in their manuscript, so I suggest making a table with comorbidities, and their prevalence.

Done, see table 1

In addition to the comorbidities mentioned by the authors, it would be good to list rheumatoid arthritis and Crohn's disease, which also occur in pediatric psoriasis.

We decided to focus our review exclusively on metabolic comorbidities.

Furthermore, since the authors in lines 51-53 mentioned cytokines it would be good to mention the difference in cytokines in children and adults in psoriasis. Pediatric psoriasis patients exhibited a distinct difference in the expression of interleukin 17 (IL-17) and IL-22 compared to that of healthy pediatric controls and adult psoriasis patients.

Done

Other corrections to be made are as follows:

  1. References in the text are not written according to the instructions. References should be written everywhere in square brackets. Done
  2. Versus (vs) should be written in italic because it comes from a Latin word, and all Latin words are written in italic Done
  3. Lines 41, 64, 65, 93, 107, 144 have too many spaces. Pass and correct the excess space in the text. Done
  4. The names of the authors are not written in italic. Only collaborators are written in italics (et al.). Please revise the whole paper. Done
  5. Also when the author's name is mentioned in the text, his name must be followed by a literary citation. Please revise the whole paper. Done
  6. Lines 66-70 connect both sentences because the first one seems incomplete on its own. Done
  7. Line 96 Pietrzak and coll should be Pietrzak et al. – please correct it. Done
  8. Line 105 missing a dot at the end of the sentence. Done
  9. Line 111 Phan and colleagues replace with Phan et al. Done
  10. Lines 113-114 rewrite the text in black Done
  11. Line 145 recommendation C - what is recommendation C, is it a mistake? Done, explained in the text.
  12. Line 150 the capital letter is missing at the beginning of the sentence.  A systemic therapy…Done

Reviewer 3 Report

This article is well written and easy to read, however, it is not offering any novel information or approach to psoriasis in the pediatric population. The authors should describe the potential implications of all those comorbidities in the treatment of psoriasis and offer their opinion on the patient management. In addition, a figure describing the interplay between psoriasis and metabolic disturbances would improve the quality of the article. Please provide more information in the abstract, it is too general. It would be useful to divide the body of text into several distinct paragraphs ie. psoriasis and obesity, psoriasis and IR, psoriasis and cardiovascular disease, etc..... 

Author Response

This article is well written and easy to read, however, it is not offering any novel information or approach to psoriasis in the pediatric population. The authors should describe the potential implications of all those comorbidities in the treatment of psoriasis and offer their opinion on the patient management.

Done, see conclusions

In addition, a figure describing the interplay between psoriasis and metabolic disturbances would improve the quality of the article.

We decided to provide a table rather than an image since we believe it is easier to consult

Please provide more information in the abstract, it is too general.

done

It would be useful to divide the body of text into several distinct paragraphs ie. psoriasis and obesity, psoriasis and IR, psoriasis and cardiovascular disease, etc..... 

done

Reviewer 4 Report

This article provides a succinct and accurate review of metabolic comorbidities in the pediatric population with psoriasis. Although some parts can be confusing (eg the beginning of the last paragraph), I consider that it deserves to be published after some small corrections.

1. What is the main question addressed by the research?

Comorbidities in pediatric psoriasis, with a focus on 
2. Do you consider the topic original or relevant in the field, and if
so, why? The topic is not strictly original, as is a review of the literature, but is a good resume of what has been published.
3. What does it add to the subject area compared with other published
material? Although it does not add anything new, it is an interesting summary to update on the publications on this subject in recent years.
4. What specific improvements could the authors consider regarding the
methodology? I think that the methodology is adequate, considering that is a small narrative review and not a comprehensive systematic review.
5. Are the conclusions consistent with the evidence and arguments
presented and do they address the main question posed? Yes
6. Are the references appropriate?

Yes, and they are recent.

Author Response

This article provides a succinct and accurate review of metabolic comorbidities in the pediatric population with psoriasis. Although some parts can be confusing (eg the beginning of the last paragraph), I consider that it deserves to be published after some small corrections.

 done

  1. What is the main question addressed by the research?

Comorbidities in pediatric psoriasis, with a focus on 
2. Do you consider the topic original or relevant in the field, and if
so, why? The topic is not strictly original, as is a review of the literature, but is a good resume of what has been published.
3. What does it add to the subject area compared with other published
material? Although it does not add anything new, it is an interesting summary to update on the publications on this subject in recent years.
4. What specific improvements could the authors consider regarding the
methodology? I think that the methodology is adequate, considering that is a small narrative review and not a comprehensive systematic review.
5. Are the conclusions consistent with the evidence and arguments
presented and do they address the main question posed? Yes
6. Are the references appropriate?

Yes, and they are recent.

Round 2

Reviewer 3 Report

I find that the authors' responses  to reviewers' comments are inadequate, inaccurate, and do not specify how and where corrections were made in the text. Each comment requires an accurate answer with the corrections listed in the manuscript.

Author Response

We apologize for the previous inaccuracy of the response, we specifically include the references of the text. 

This article is well written and easy to read, however, it is not offering any novel information or approach to psoriasis in the pediatric population. The authors should describe the potential implications of all those comorbidities in the treatment of psoriasis and offer their opinion on the patient management.

Done

"In conclusion, dermatologists should be aware of metabolic comorbidities in children with psoriasis, modulating the therapeutic approach according to the patient's clinical condition. In our opinion, topical therapy should be considered as the first line treatment in cases of limited disease (table 1). When systemic therapy is required, the effects of drugs on metabolic comorbidities must be evaluated. Specifically, BMI screening, fasting serum glucose, universal lipid screening and hypertension screening should be done for psoriatic children at risk of overweight or obesity, implementing lifestyle modifications, with the support of a nutritionist, to prevent their progression.

Among conventional systemic drugs, acitretin, cyclosporine and methotrexate should be avoided in cases of obesity and dyslipidemia, the latter because it is potentially associated with an increased hepatotoxic risk in patients with obesity and diabetes. Fumaric acid esters and apremilast do not appear to have contraindications related to metabolic complications, but their use is still limited in the pediatric population. [31]

According to our opinion, we suggest considering biological therapy as a first-line alternative in the event of topical therapy or phototherapy failure in pediatric psoriatic patient with high risk of metabolic comorbidities. In particular, TNF alpha inhibitors (etanercept and adalimumab), IL-17 inhibitors (ixekizumab, secukinumab) and IL 12/23 inhibitors (ustekinumab) are available for pediatric use [31] and do not appear to have any interactions from the point of view of psoriasis metabolic complication"

In addition, a figure describing the interplay between psoriasis and metabolic disturbances would improve the quality of the article.

We decided to provide a table rather than an image since we believe it is easier to consult. See Table 1 in the text.

Please provide more information in the abstract, it is too general.

done

"Here, we summarize and discuss, in a narrative review, current knowledge about the metabolic comorbidities in psoriatic children. Obesity, insulin resistance, diabetes, cardiovascular disease and dyslipidemia are identified as the main comorbidities in psoriatic children. In conclusion, dermatologists should be aware of metabolic comorbidities in children with psoriasis, modulating the therapeutic approach according to the patient's clinical condition."

It would be useful to divide the body of text into several distinct paragraphs ie. psoriasis and obesity, psoriasis and IR, psoriasis and cardiovascular disease, etc.....

done, see line 41, 77, 109, 150, and 179